# Aligning Model and Macaque Inferior Temporal Cortex Representations Improves Model-to-Human Behavioral Alignment and Adversarial Robustness

Joel Dapello[*,1,2,3], Kohitij Kar[*,1,2,4,6],
Martin Schrimpf[1,2,4], Robert Geary[1,2,3], Michael Ferguson[1,2,4] David D. Cox[5], James J. DiCarlo[1,2,4]
[1]Department of Brain and Cognitive Sciences, MIT, Cambridge, MA02139
[2]McGovern Institute for Brain Research, MIT, Cambridge, MA02139
[3]School of Engineering and Applied Sciences, Harvard University, Cambridge, MA02139
[4]Center for Brains, Minds and Machines, MIT, Cambridge, MA02139
[5]MIT-IBM Watson AI Lab
[6] Department of Biology, Centre for Vision Research at York University, Toronto, CA
dapello@mit.edu      kohitij@mit.edu

## Abstract

While some state-of-the-art artificial neural network systems in computer vision are strikingly accurate models of the corresponding primate visual processing, there are still many discrepancies between these models and the behavior of primates on object recognition tasks. Many current models suffer from extreme sensitivity to adversarial attacks and often do not align well with the image-by-image behavioral error patterns observed in humans. Previous research has provided strong evidence that primate object recognition behavior can be very accurately predicted by neural population activity in the inferior temporal (IT) cortex, a brain area in the late stages of the visual processing hierarchy. Therefore, here we directly test whether making the late stage representations of models more similar to that of macaque IT produces new models that exhibit more robust, primate-like behavior. We collected a dataset of chronic, large-scale multi-electrode recordings across the IT cortex in six non-human primates (rhesus macaques). We then use these data to fine-tune (end-to-end) the model "IT" representations such that they are more aligned with the biological IT representations, while preserving accuracy on object recognition tasks. We generate a cohort of models with a range of IT similarity scores validated on held-out animals across two image sets with distinct statistics. Across a battery of optimization conditions, we observed a strong correlation between the models' IT-likeness and alignment with human behavior, as well as an increase in its adversarial robustness. We further assessed the limitations of this approach and find that the improvements in behavioral alignment and adversarial robustness generalize across different image statistics, but not to object categories outside of those covered in our IT training set. Taken together, our results demonstrate that building models that are more aligned with the primate brain leads to more robust and human-like behavior, and call for larger neural data-sets to further augment these gains. Code, models, and data are available at https://github.com/dapello/braintree.

## 1 Introduction and Related Work

Object recognition models have made incredible strides in the last ten years, (Krizhevsky et al., 2012; Szegedy et al., 2014; Simonyan and Zisserman, 2014; He et al., 2015b; Dosovitskiy et al., 2020; Liu et al., 2022) even surpassing human performance in some benchmarks (He et al., 2015a). While some of these models bear remarkable resemblance to the primate visual system (Daniel L. Yamins, 2013;

---

[*]These authors contributed equally to this work.

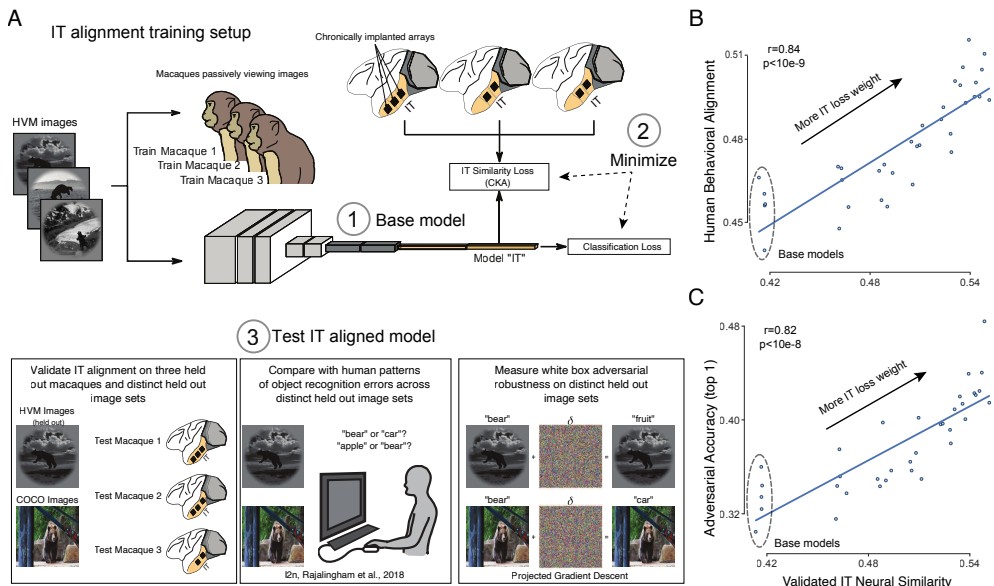

Figure 1: **Aligning model IT representations with primate IT representations improves behavioral alignment and improves adversarial robustness. A**) A set of naturalistic images, each containing one of eight different object classes are shown to a CNN and also to three different primate subjects with implanted multi-electrode arrays recording from the Inferior Temporal (IT) cortex. (1) A Base model (ImageNet pre-trained CORnet-S) is fine-tuned using stochastic gradient descent to (2) minimize the classification loss with respect to the ground truth object in each image while also minimizing a representational similarity loss (CKA) that encourages the model's IT representation to be more like those measured in the (pooled) primate subjects. (3) The resultant IT aligned models are then frozen and each tested in three ways. First, model IT representations are evaluated for similarity to biological IT representation (CKA metric) using neural data obtained from new primate subjects – we refer to the split-trial reliability ceiled average across all held out macaques and both image sets as "Validated IT neural similarity". Second, model output behavioral error patterns are assessed for alignment with human behavioral error patterns at the resolution of individual images (i2n, see Methods). Third, model behavioral output is evaluated for its robustness to white box adversarial attacks using an $L_\infty$ norm projected gradient descent attack. All three tests are carried out with: (i) new images within the IT-alignment training domain (held out HVM images; see Methods) and (ii) new images with novel image statistics (natural COCO images; see Methods), and those empirical results are tracked separately. **B**) We find that this IT-alignment procedure produced gains in validated IT neural similarity relative to base models on both data sets, and that these gains led to improvement in human behavioral alignment. n=30 models are shown, resulting from training at six different relative weightings of the IT neural similarity loss, each from five base models that derived from five random seeds. **C**) We also find that these same IT-alignment gains resulted in increased adversarial accuracy (PGD $L_\infty$, $\epsilon = 1/1020$) on the same model set as in **B**. Base models trained only for ImageNet and HVM image classification are circled in grey.

Khaligh-Razavi and Kriegeskorte, 2014; Schrimpf et al., 2018; 2020), there remain a number of important discrepancies. In particular, the output behavior of current models, while coarsely aligned with primate object confusion patterns, does not fully match primate error patterns on individual images (Rajalingham et al., 2018; Geirhos et al., 2021). In addition, these same models can be easily fooled by adversarial attacks – targeted pixel-level perturbations intentionally designed to cause the model to produce the wrong output(Szegedy et al., 2013; Carlini and Wagner, 2016; Chen et al., 2017; Rony et al., 2018; Brendel et al., 2019), whereas primate behavior is thought to be more robust to these kinds of attacks. This is an important unsolved problem in engineering artificial intelligence systems; the deviance between model and human behavior has been studied extensively in the machine learning community, often from the perspective of safety in real-world deployment of computer vision systems (Das et al., 2017; Liu et al., 2017; Xu et al., 2017; Madry et al., 2017; Song et al., 2017; Dhillon et al., 2018; Buckman et al., 2018; Guo et al., 2018; Michaelis et al., 2019). From a neuroscience perspective, behavioral differences like these point to different underlying mechanisms

and feature representations used for object recognition between the artificial and biological systems, meaning that our scientific understanding of the mechanisms of visual behavior remains incomplete.

Incorporating neurophysiological constraints into models to make them behave more in line with primate visual behavior is an active field of research (Marblestone et al., 2016; Lotter et al., 2016; Nayebi and Ganguli, 2017; Guerguiev et al., 2017; Hassabis et al., 2017; Lindsay and Miller, 2018; Tang et al., 2018; Kar et al., 2019; Kubilius et al., 2019; Li et al., 2019; Hasani et al., 2019; Sinz et al., 2019; Zador, 2019; Geiger et al., 2022). Previously, Dapello et al. (2020) demonstrated that convolutional neural network (CNN) models with early visual representations that are more functionally aligned with the early representations of primate visual processing tended to be more robust to adversarial attacks. This correlational observation was turned into a causal test, by simulating a primary visual cortex at the front of CNNs, which was indeed found to improve performance across a range of white box adversarial attacks and common image corruptions. Likewise, several recent studies have demonstrated that training models to classify images while also predicting (Safarani et al., 2021) or having similar representations (Federer et al., 2020) to early visual processing regions of primates, or even mice (Li et al., 2019), has a positive effect on generalization and robustness to adversarial attacks and common image corruptions.

However, no research to date has investigated the effects of incorporating biological knowledge of the neural representations in the IT cortex – a late stage visual processing region of the primate ventral stream, which critically supports primate visual object recognition (DiCarlo et al., 2012; Majaj et al., 2015). Here, we developed a method to align the late layer "IT representations" of a base object recognition model (CORnet-S (Kubilius et al., 2019) pre-trained on ImageNet (Deng et al., 2009) and naturalistic, grey-scale "HVM" images (Majaj et al., 2015)) to the biological IT representation while the model continues to be optimized to perform classification of the dominant object in each image. Using neural recordings performed across the IT cortex of six rhesus macaque monkeys divided into three training animals and three held-out testing animals for validation, we generate a suite of models under a variety of different optimization conditions and measure their IT alignment on held out animals, their alignment with human behavior, and their robustness to a range of adversarial attacks, in all cases on at least two image sets with distinct statistics as shown in figure 1.

We report three novel findings:

1. Our method robustly improves IT representational similarity of models to brains even when measured on new animals and new images.
2. We find that gains in model IT-likeness lead to gains in human behavioral alignment.
3. Likewise we find that improved IT-likeness leads to increased adversarial robustness.

Interestingly, we observe that adversarial training improves robustness but does not significantly increase IT similarity or human behavioral alignment. Finally, while probing the limits of our current IT-alignment procedure, we observed that the improvements in IT similarity, behavioral alignment, and adversarial robustness generalized to images with different image statistics than those in the IT training set (from naturalistic gray scale images to full color natural images) but only for object categories that were part of the original IT training set and not for held-out object categories.

## 2 DATA AND METHODS

Here we describe the neural and behavioral data collection, the training and testing methods used for aligning model representations with IT representations, and the methods for assessing behavioral alignment and adversarial robustness.

### 2.1 IMAGE SETS

High-quality synthetic "naturalistic" images of single objects (HVM images) were generated using free ray-tracing software (http://www.povray.org), similar to (Majaj et al., 2015). Each image consisted of a 2D projection of a 3D model (purchased from Dosch Design and TurboSquid) added to a random natural background. The ten objects chosen were bear, elephant, face, apple, car, dog, chair, plane, bird and zebra. By varying six viewing parameters, we explored three types of identity while preserving object variation, position (x and y), rotation (x, y, and z), and size. All images

were achromatic with a native resolution of 256 × 256 pixels. Additionally, natural microsoft COCO images (photographs) pertaining to the 10 nouns, were download from http://cocodataset.org (Lin et al., 2014). Each image was resized (not cropped) to 256 x 256 x 3 pixel size and presented within the central 8 deg.

## 2.2 PRIMATE NEURAL DATA COLLECTION AND PROCESSING

We surgically implanted each monkey with a head post under aseptic conditions. We recorded neural activity using two or three micro-electrode arrays (Utah arrays; Blackrock Microsystems) implanted in IT cortex. A total of 96 electrodes were connected per array (grid arrangement, 400 um spacing, 4mm x 4mm span of each array). Array placement was guided by the sulcus pattern, which was visible during the surgery. The electrodes were accessed through a percutaneous connector that allowed simultaneous recording from all 96 electrodes from each array. All surgical and animal procedures were performed in accordance with National Institutes of Health guidelines and the Massachusetts Institute of Technology Committee on Animal Care. For information on the neural recording quality metrics per site, see supplemental section A.1.

During each daily recording session, band-pass filtered (0.1 Hz to 10 kHz) neural activity was recorded continuously at a sampling rate of 20 kHz using Intan Recording Controllers (Intan Technologies, LLC). The majority of the data presented here were based on multiunit activity. We detected the multiunit spikes after the raw voltage data were collected. A multiunit spike event was defined as the threshold crossing when voltage (falling edge) deviated by more than three times the standard deviation of the raw voltage values. Our array placements allowed us to sample neural sites from different parts of IT, along the posterior to anterior axis. However, for all the analyses, we did not consider the specific spatial location of the site, and treated each site as a random sample from a pooled IT population. For information on the neural recording quality metrics, see supplemental section A.1.

***Behavioral state during neural data collection*** All neural response data were obtained during a passive viewing task. In this task, monkeys fixated a white square dot (0.2°) for 300 ms to initiate a trial. We then presented a sequence of 5 to 10 images, each ON for 100 ms followed by a 100 ms gray blank screen. This was followed by a water reward and an inter trial interval of 500 ms, followed by the next sequence. Trials were aborted if gaze was not held within ±2° of the central fixation dot during any point. Each neural site's response to each image was taken as the mean rate during a time window of 70-170ms following image onset, a window that has been previously chosen to align with the visually-driven latency of IT neurons and their quantitative relationship to object classification behavior as in Majaj et al. (2015).

## 2.3 HUMAN BEHAVIORAL DATA COLLECTION

We measured human behavior (from 88 subjects) using the online Amazon MTurk platform which enables efficient collection of large-scale psychophysical data from crowd-sourced "human intelligence tasks" (HITs). The reliability of the online MTurk platform has been validated by comparing results obtained from online and in-lab psychophysical experiments (Majaj et al., 2015; Rajalingham et al., 2015). Each trial started with a 100 ms presentation of the sample image (one our of 1320 images). This was followed by a blank gray screen for 100 ms; followed by a choice screen with the target and distractor objects, similar to (Rajalingham et al., 2018). The subjects indicated their choice by touching the screen or clicking the mouse over the target object. Each subjects saw an image only once. We collected the data such that, there were 80 unique subject responses per image, with varied distractor objects. Prior work has shown that human and macaque behavioral patterns are nearly identical, even at the image grain (Rajalingham et al., 2018). For information on the human behavioral data collection, see supplemental section A.2.

## 2.4 ALIGNING MODEL REPRESENTATIONS WITH MACAQUE IT REPRESENTATIONS

In order to align neural network model representations with primate IT representations while performing classification, we use a multi-loss formulation similar to that used in Li et al. (2019) and Federer

et al. (2020). Starting with an ImageNet (Deng et al., 2009) pre-trained[1] CORnet-S model (Kubilius et al., 2019), we used stochastic gradient descent (SGD) on all model weights to jointly minimize a standard categorical cross entropy loss on model predictions of ImageNet labels (maintained from model pre-training, for stability), HVM image labels, and a centered kernel alignment (CKA) based loss penalizing the "IT" layer of CORnet-S for having representations not aligned with primate IT representations of the HVM images. CORnet-S was selected because it already has a clearly defined layer committed to region IT, close to the final linear readout of the network, but otherwise our procedure is compatible with any neural network architecture. Meanwhile CKA, a measure of linear subspace alignment, was selected as a representational similarity measure. CKA has ideal properties such as invariance to isotropic scaling and orthonormal transformations which do not matter from the perspective of a linear readout, but sensitivity to arbitrary linear transformations (Kornblith et al., 2019) which could lead to differences from a linear readout as well as allow the network to hide representations useful for image classification but not present within primate IT. CKA ranges from 0, indicating completely non-overlapping subspaces, to 1, indicating completely aligned subspaces. We found that our best neural alignment results came from minimizing the neural similarity loss function $log(1 - CKA(X, Y))$, where $X \in \mathbb{R}^{n \times p_1}$ and $Y \in \mathbb{R}^{n \times p_2}$ denote two column centered activation matrices with generated by showing $n$ example images and recording $p_1$ and $p_2$ neurons from the IT layer of CORnet-S and macaque IT recordings respectively. The macaque neural activation matrices were generated by averaging over approximately 50 trials per image and over a 70-170 millisecond time window following image presentation. An illustration of our setup is shown in figure 1A.

## 2.5 TRAINING AND TESTING CONDITIONS

In all reported experiments, model IT representational similarity training was performed on 2880 grey-scale naturalistic HVM image representations consisting of 188 active neural sites collated from the three training set macaques for 1200 epochs. We use a batch size of 128, meaning the CKA loss computed for a random set of 128 representations for each gradient step. In order to create models with a variety of different final neural alignment scores, we add random probability $1 - p$ of dropping the IT alignment gradients and create six different sets (5 random seeds for each set) of neurally aligned models with $p \in [0, 1/32, 1/16, 1/8, 1/4, 1/2, 1]$. For example, the set with $p = 0$ drops all of the IT alignment gradients and thus has no improved IT alignment over the base model, while the set with $p = 1$ always includes the IT alignment gradients and similarly achieves the highest IT alignment scores (see figure 2). We also introduce a small amount of data augmentation including the physical equivalent of 0.5 degrees of jitter the vertical and horizontal position of the images, 0.5 degrees of rotational jitter, and +/- 0.1 degrees of scaling jitter, assuming our model has an 8 degree field of view. These augmentations were selected to simulate natural viewing conditions.

Model IT representational similarity testing was performed on a total of three held out monkeys: Monkey 1 (280 neural sites) and monkey 2 (144 neural sites) on 320 held out HVM images with statistics similar to the training distribution, and monkey 1 (237 neural sites) and monkey 3 (106 active neural sites) on 200 full color natural COCO images with different statistics than those used during training. Additional model training information can be found in supplemental section B.

For performing white box adversarial attacks, we used untargeted projected gradient descent (PGD) (Madry et al., 2017) with $L_\infty$ and $L_2$ norm constraints. Further details are given in supplemental section B.

## 2.6 BEHAVIORAL BENCHMARKS

To characterize the behavior of the visual system, we have used an image-level behavioral metric, *i2n* (Rajalingham et al., 2018). The behavioral metric computes a pattern of unbiased behavioral performances, using a sensitivity index: $d' = Z(HitRate) - Z(FalseAlarmRate)$, where $Z$ is the inverse of the cumulative Gaussian distribution. The HitRates for i2n are the accuracies of the subjects when a specific image is shown and the choices include the target object (i.e., the object present in the image) and one other specific distractor object. So for every distractor-target pair we get a different i2n entry. A detailed description of how to compute i2n can be also found at Rajalingham

---

[1] the pre-trained version was selected as a starting point because of the relatively small number of training samples in our dataset (Riedel, 2022).

et al. (2018). The i2n behavioral benchmark was computed using the Brain-Score implementation of the i2n metric (Schrimpf et al., 2018).

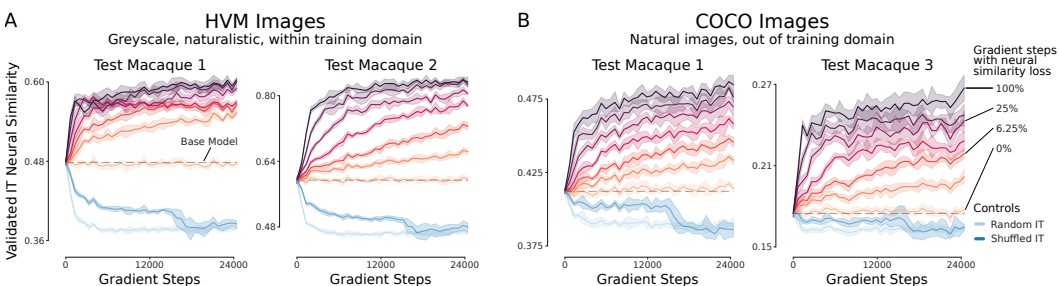

Figure 2: **IT alignment training leads to improved IT representational similarity on held out animals and held out images across two image sets with different statistics. A**) IT neural similarity scores (CKA, normalized by split-half trial reliability) for held out but within domain HVM images vs gradient steps is shown for two held out monkeys across seven different neural similarity loss gradient dropout rates (the darkest trace receives neural similarity loss gradients at 100% of gradient steps, while in the lightest trace neural similarity loss gradients are dropped at every step). Two control conditions are also shown: optimizing model IT toward a random Gaussian target IT matrix (random, blue) and toward an image-shuffled target IT matrix (shuffle, orange). **B**) Like **A** but for natural COCO images out of domain with respect to the training set. Grey dashed line on each plot shows the base model score for models pre-trained on ImageNet and HVM image labels with no IT representational similarity loss, which the model set with 0% of IT similarity loss gradients does not deviate significantly from. Error bars are bootstrapped confidence intervals for 5 training seeds.

## 3  RESULTS

Does aligning late stage model representations with primate IT representations lead to improvements in alignment with image-by-image patterns of human behavior or improvements in white box adversarial robustness? We start by testing if our method can generate models that are truly more IT-like by validating on held out animals and images, as this has not been previously attempted and is not guaranteed to work given the sampling limitations of neural recording experiments. We then proceed to analyze how these IT-aligned models fair on several human behavioral alignment benchmarks and a diverse set of white box adversarial attacks.

### 3.1  DIRECT FITTING TO IT NEURAL DATA IMPROVES IT-LIKENESS OF MODELS ACROSS HELD OUT ANIMALS AND IMAGE SETS

First, we investigated how well our IT alignment optimization procedure generalizes to IT neural similarity measurements (CKA) for two held out test monkeys on 320 held out HVM images (similar image statistics as the training set). Figure 2A shows the ceiled IT neural similarity scores for both test animals across different neural similarity loss gradient dropout rates ($p \in [0, 1/32, 1/16, 1/8, 1/4, 1/2, 1]$; the model marked 100% sees IT similarity loss gradients at every step, where as the model marked 0% never sees IT similarity loss gradients) as well as models optimized to classify HVM images while fitting a random Gaussian target activation matrix, or an image shuffled target activation matrix which has the same first and second order statistics as the true IT activation matrix, but scrambled image information. For both animals, we see a significant positive shift from the unfitted model (neural loss weight of 0.0), with higher relative neural loss weights generally leading to higher IT neural similarity scores. Meanwhile, both of the control conditions cause models to become less IT like, to a significant degree.

We next investigated how well our procedure generalizes from the grey-scale naturalistic HVM images to full color, natural images from COCO. Figure 2B shows the same model optimization conditions as before, but now on two unseen animal IT representations of COCO images. Like in 2A although to a lesser absolute degree, we see improvements relative to the baseline in IT neural similarity as function of the neural loss weight, and controls generally decreasing in IT neural similarity. From

this, we conclude that our IT alignment procedure is able to improve IT-likeness in our models even in held out animals and across two image sets with distinct statistics.

## 3.2 INCREASED BEHAVIORAL ALIGNMENT IN MODELS THAT BETTER MATCH MACAQUE IT

Next, we investigated how single image level classification error patterns correlate between humans and IT aligned models. To get a big picture view, we take all of the optimization conditions and validation epochs generated in figure 2A while models are training and compare IT neural similarity on the HVM test set (averaged over held out animals) with human behavioral alignment on the HVM test set. As shown in figure 3A, this analysis reveals a broad, though not linear correlation between IT neural similarity and behavioral alignment. Interestingly, we observe that the slope is at its steepest when IT neural similarity is at the highest values, suggesting that if an even higher degree of IT-alignment might result in greater increases in behavioral alignment. We also investigated whether these trends persist when we exclude the optimization on object labels from the HVM images and only optimize for IT neural similarity. To do so, we train the models on all previous conditions but without the HVM object-label loss. As shown in 3, the overall shape of the trend remains quite similar, though the absolute behavioral alignment shifts downward, indicating that the label information during training helps on the behavioral task, but is not required for the trend to hold. In figure 3B, we perform the same set of measurements but now focusing on the COCO image set. Consistent with the observation on COCO IT neural similarity, the behavioral alignment trend transfers to the COCO image set although the absolute magnitude of the improvements are less.

Finally, using the Brain-Score platform (Schrimpf et al., 2018), we benchmark our models against publicly available human behavioral data from the Objectome image set Rajalingham et al. (2018) which has similar image statistics to our HVM IT fitting set (with a total of 24 object categories, only four of which overlap with the training set). As demonstrated in figure 4C, when the Objectome data are filtered down to just the four overlapping categories, our most IT similar models are again the most behaviorally aligned, well above the unfit baseline and control conditions, which remain close to the floor for much of the plot. However, As shown in figure 3D, when considering all 24 object categories in the Objectome dataset, we see that the trend of increasing human behavioral alignment does not hold and our models actually begin to fair worse in terms of human behavioral alignment at higher levels of IT neural similarity. As shown in figure supp A.1, using a linear probe to assess image class information content (measured by classification accuracy on held out representations) reveals that these models are losing class information content for the Objectome image set, which drives the decrease in behavioral alignment, as the model makes more mistakes overall than a human. Similarly, a linear probe analysis reveals minimal loss in class information in the overlapping categories. Thus, we observe that while our method leads to increased human behavioral alignment across different image statistics, it does not currently lead to improved alignment on unseen object categories.

## 3.3 INCREASED ADVERSARIAL ROBUSTNESS IN MODELS THAT BETTER MATCH MACAQUE IT

Finally, we evaluate our models on an array of white box adversarial attacks, to assess if models with higher IT neural similarity scores also have increased adversarial robustness. Like before, we start with a big picture analysis where we consider every evaluation epoch for all optimization conditions considered in figure 2. Again, as demonstrated in figures 4A and 4B, for both HVM images and COCO images, there is a broad though not entirely linear correlation between IT neural similarity and adversarial robustness to PGD $L_\infty$ $\epsilon = 1/1020$ attacks. Like in the analysis of behavioral alignment, we also see a higher slope on the right side of the plots, where IT neural similarity is the highest, suggesting further improvements could be had if models were pushed to be more IT aligned.

In order to get a better sense of the gains in robustness, we measured the adversarial strength accuracy curves for models only trained with HVM image labels, models trained with HVM image labels and IT neural representations, and models adversarially trained on HVM labels (PGD $L_\infty$, $\epsilon = 4/255$). Figure 5A shows that on held-out HVM images, IT aligned models have increased accuracy across a range of $\epsilon$ values for both $L_\infty$ and $L_2$ norms, though less so than models with explicit adversarial training. However, as shown in figure 5 the same analysis on COCO images demonstrates that adversarial robustness in the IT aligned networks generalizes significantly better on unseen image statistics than the adversarially trained models, which lose clean accuracy on COCO images.

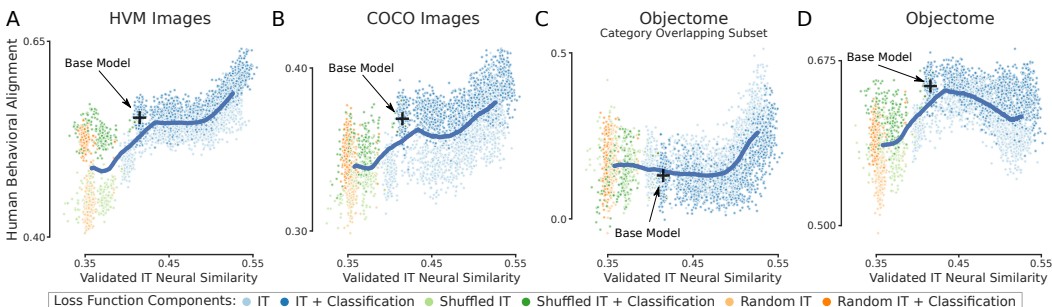

Figure 3: **IT neural similarity correlates with behavioral alignment across a variety of optimization conditions and unseen image statistics but not on unseen object categories. A**) Held out animal and image IT neural similarity is plotted against human behavioral alignment on the HVM image set at every validation epoch for all neural loss weight conditions, random Gaussian IT target matrix conditions, and image shuffled IT target matrix conditions, in each case with or with and without image classification loss. **B**) and **C**) Like in **A** but for the COCO image set and the Objectome image set Rajalingham et al. (2018) filtered to overlapping categories with the IT training set. **D**) The behavioral alignment for the full Objectome image set with 20 categories not covered in the IT training is not improved by the IT-alignment procedure and data used here. In all plots, the black cross represents the average base model position, and the heavy blue line is a sliding X, Y average of all conditions merely to visually highlight trends. Five seeds for each condition are plotted.

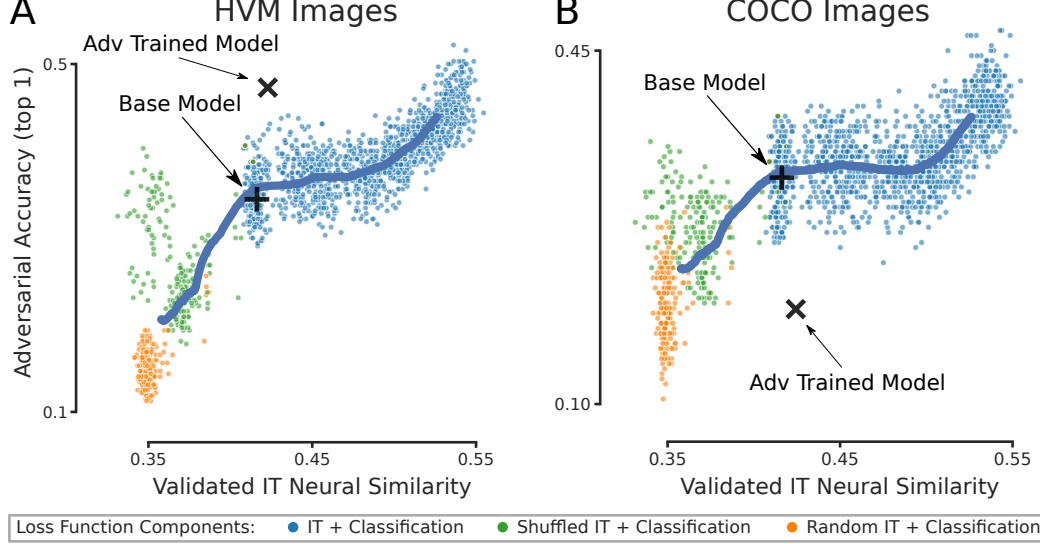

Figure 4: **IT neural similarity correlates with improved white box adversarial robustness. A**) held out animal and image IT neural similarity is plotted against white box adversarial accuracy (PGD $L_\infty$ $\epsilon = 1/1020$) on the HVM image set measured across multiple training time points for all neural loss ratio conditions, random Gaussian IT target matrix conditions, and image shuffled IT target matrix conditions. **B**) Like in **A** but for COCO images. In both plots, the black cross represents the average base model position, the black X marks a CORnet-S adversarially trained on HVM images, and the heavy blue line is a sliding X, Y average of all conditions merely to visually highlight trends. Five seeds for each condition are plotted.

Last, we tested the IT neural similarity of our HVM image adversarially trained models and find that they do not follow the general correlation shown in 4 for IT aligned models vs adversarial accuracy. Interestingly, the adversarially trained models are slightly more similar to IT than standard models, but significantly higher than standard models on HVM adversarial accuracy and significantly lower on COCO adversarial accuracy. We take this to indicate that there are multiple possible ways to become robust to adversarial attacks, and that adversarial training does not in general induce the same representations as IT alignment.

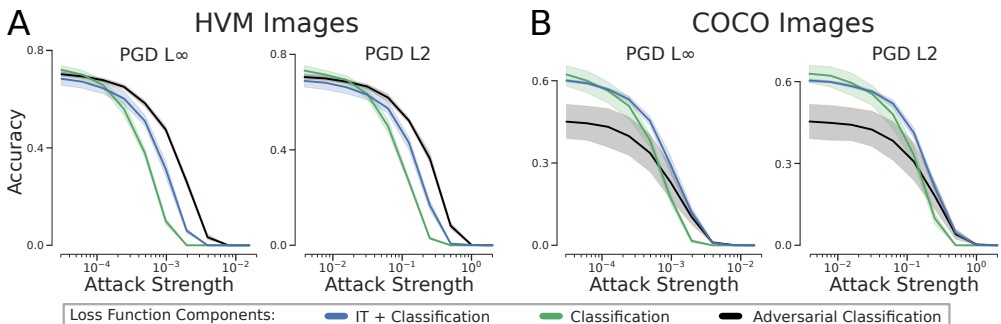

Figure 5: **IT aligned models are more robust than standard models in and out of domain, and more robust than adversarially trained models in out of domain conditions. A**) PGD $L_\infty$ and $L_2$ strength accuracy curves on HVM images for standard trained networks (green) IT aligned networks (blue) and networks adversarially trained (PGD $L_\infty$ $\epsilon = 4/255$) on the IT fitting image labels (orange). **B**) Like in **A** but for COCO images. Error shading represents bootstrapped 95% confidence intervals over five training seeds.

## 4 DISCUSSION

Building on prior research in constraining visual object recognition models with early stage visual representations (Li et al., 2019; Dapello et al., 2020; Federer et al., 2020; Safarani et al., 2021), we report here that it is possible to better align the late stage "IT representations" of an object recognition model with the corresponding primate IT representations, and that this improved IT alignment leads to increased human level behavioral alignment and increased adversarial robustness. In particular, the results show that 1) the method used here is able to develop better neuroscientific models by improving IT alignment in object recognition models even on held out animals and image statistics not seen by the model during the IT neural alignment training procedure, 2) models that are more aligned with macaque IT also have better alignment with human behavioral error patterns across unseen (not shown during training) image statistics but not for unseen object categories, and 3) models more aligned with macaque IT are more robust to adversarial attacks even on unseen image statistics. Interestingly however, we observed that being more adversarially robust (through adversarial training) does not lead to significantly more IT neural similarity.

These empirical observations raise a number of important questions for future research. While there are clear gains in robustness from our procedure, we note that the overall magnitude is relatively small. How much adversarial robustness could we expect to gain, if we perfectly fit IT? This question hinges on how adversarially robust primate behavior really is, an active area of research (Guo et al., 2022; Elsayed et al., 2018; Yuan et al., 2020). Guo et al. (2022) is particularly interesting with respect to our work – while they find that individual neurons in IT are not particularly robust when compared to individual neurons in adversarially trained networks, our work here indicates that population geometry, not individual neuronal sensitivity, might play a critical role in robustness. We find it intriguing that aligning IT representations in our models to empirically measured macaque IT responses has no effect or even a negative effect on behavioral alignment for objects not present in the IT fitting image-set, a noteworthy limitation in our approach. We speculate that this is due to the small range of categories covered in our IT training set, which limits the span of neural representational space that those experiments were able to sample. In that regard, it would be informative to get a sense of the scaling laws (Kaplan et al., 2020) for how much neural data (in terms of images, neurons, trials, or object categories) needs to be absorbed into a model before it behaves in a truly general more human like fashion for any instance of image categories or statistics. Other avenues for further exploration include comparisons of behavioral alignment on a more diverse panel of benchmarks Bowers et al. (2022), different alignment metrics to optimize, such as deep canonical correlationPirlot et al. (2022), or including representation stochasticity as in Dapello et al. (2020). Overall, our results provide further support for the framework of constraining and optimizing models with empirical data from the primate brain to make them more robust and well aligned with human behavior (Sinz et al., 2019).

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
