# OpenReview forum: "Aligning Model and Macaque Inferior Temporal Cortex Representations Improves Model-to-Human Behavioral Alignment and Adversarial Robustness"
_ICLR.cc/2023/Conference — ICLR 2023 notable top 5%_

### Official Review · Reviewer_Bkvv · 2022-10-23

**Confidence:** 4
**Correctness:** 4
**Technical Novelty And Significance:** 3
**Empirical Novelty And Significance:** 3
**Recommendation:** 8

**Clarity, Quality, Novelty And Reproducibility:**

The paper is clearly written, and the work is of high quality and
addresses an interesting quesion with novel data and reasonably novel approaches. The
experimental and analysis procedures are described in sufficient
detail, but I could not find links to data (human or monkey) or code.

**Strength And Weaknesses:**

## Strengths
1. The paper contains a wealth of experimental data, including human
   psychophysics and monkey electrophysiology. The number of monkeys
   is large for this type of study (6), which allows to test
   similarity metrics on held-out animals.
2. The evidence supporting the results is generally convincing.
3. The paper clearly underscores one limitation of the current
   approach/set of results, by pointing out how the increase in
   behavioral match between humans and network with increasing
   representational alignment does not hold for object categories
   not included in the training set. Some possible ideas for a way
   forward on this issue are proposed in the discussion.

## Weaknesses
1. I could not find a link to the experimental data collected for this
   work, or a promise to publish such data upon acceptance.
2. Section 3.2 states that "the label information during training
   helps on the behavioral task, but is not required for the trend [of
   better representation alignment improving behavioral match between
   humans and deep nets] to hold". This is correct, but in my opinion
   this underplays the fact that label information is still used for
   the pre-training of the network. Perhaps this passage could be
   rephrased to remind the reader that label information is
   nevertheless still available from the pre-training, even when it's
   not included in the fine-tuning process.
3. If I have understood correctly, the monkey data was recorded
   specifically for this work. However, the phrasing around this fact
   in the supplement is somewhat confusing. Please clarify, stating
   explicitly that the data previously recorded from these monkeys for
   various research objective is different data that was not used in
   this work.
4. I urge the authors to consider raising their compensation for human
   subjects in future experiments. Psychophysics subjects were paid 4 USD an hour in this work (Supplementary Material). This seems very low
   considering that the experimenters are based in Massachusetts
   (based on the information in the paper), and the minimum hourly
   wage is 7.25 USD federally and 14.25 USD in MA.

**Summary Of The Paper:**

In this paper, a deep convolutional neural network pretrained on an
object classification task is fine-tuned tp encourage
representation alignment of a specific layer with recordings from IT
cortex in macaques. The behavior of the trained network is
analyzed, finding that representational alignment with IT correlates
with a closer behavioral similarity to human behavioral patterns on
the task, and with better resilience to adversarial attacks.

**Summary Of The Review:**

This is a solid paper, which will be of interest to the ICLR
community. However, the reproducibility of the results and the
possibility for other groups to build upon the ideas presented here
could be hindered by lack of access to the experimental data collected
for this work.

---

> ### Author Response · Authors · 2022-11-11
> **Anonymous code and data release, and clarification on primate ephys data**
>
> Thank you Reviewer Bkvv for your detailed critique of our submission. We were happy you found our paper clearly written, the work to be of high quality, and the content to be of general interest to ICLR. Below we address your concerns.
>
> First and foremost, regarding Weakness 1) we fully agree and have made an anonymous preliminary release of our source code and data [here](https://anonymous.4open.science/r/IT-Alignment-7820/README.md) and we will polish and provide further documentation of this repository with our final published paper. Regarding Weakness 2) we agree with the reviewer and will update the text with their suggestion to emphasize that label information is available from ImageNet pre-training, so as not to imply that it is solely arising from IT alignment. Regarding Weakness 3) we will change the wording in our paper to clarify the following. The primate electrophysiology data used in our study has indeed been used in previous research papers, as cited in our paper and supplement. For this paper we curated and aligned these recording datasets which in previous works were collected with different scientific objectives and not used in the same way as in this work. Our current work required neural responses on the same images, across many monkeys (n=6). The typical standard in non-human primate (NHP) research is to use two animals. Therefore, we had selected a set of images that were repeated across all 6 animals for addressing the questions in this project. However, to optimize our scientific output, these individual animals have also been used for other studies (as cited appropriately in the article). Finally, we accept the reviewer’s critique for Weakness 4), and we agree to increase the base payment in our future work.
>
> Again, we thank the reviewer for highlighting these points, and we think addressing them will help make our paper more clear.

---

### Official Review · Reviewer_LQZr · 2022-10-24

**Confidence:** 4
**Correctness:** 4
**Technical Novelty And Significance:** 4
**Empirical Novelty And Significance:** 4
**Recommendation:** 8

**Clarity, Quality, Novelty And Reproducibility:**

Overall I think it is a great paper that presents a simple idea in a clear manner and is well executed. As far as I can tell, demonstrating that aligning CNN and IT responses leads to more human-like behavior of the net is novel, while previous works have hinted at increased robustness by aligning CNN representations to the brain, albeit at earlier stages of the visual system.

Reproducibility should not be an issue, since the experimental manipulations are straightforward and well documented. However, unfortunately a statement regarding availability of code is missing.


**Strength And Weaknesses:**

### Strengths

 + Simple idea well executed
 + Improved alignment of CNNs with human perception
 + Improved adversarial robustness
 + Paper is well written and easy to read


### Weaknesses

 1. Unclear to what extent this result generalizes to other network architectures
 1. Improvements are relatively small


**Summary Of The Paper:**

The authors show that aligning the internal representations of CNNs trained on image classification with macaque inferotemporal cortex renders those nets' image-to-image error patterns more aligned with humans and improves their adversarial robustness.


**Summary Of The Review:**

Great paper that presents a simple idea in a clear manner and is well executed.

---

> ### Author Response · Authors · 2022-11-11
> **Thank you; we release our code anonymously**
>
> Thank you Reviewer LQZr for your positive feedback in reviewing our work; we were happy to see that you found our research to be well executed and presented clearly.
>
> Regarding the weaknesses found, while we see no reason why our method should not generalize to other architectures, we do not currently have evidence to demonstrate generalization and we accept the reviewer’s critique. Likewise, we agree that the improvements are relatively small. We hope that by incorporating more primate data in the future, we can see a larger effect size, but again for the purposes of this paper we agree with your critiques. We propose to add a note of these limitations to the discussion. Finally, we have made an anonymous preliminary release of our code and source data [here](https://anonymous.4open.science/r/IT-Alignment-7820/README.md), which will be polished and further documented for the final release of our paper.

---

### Official Review · Reviewer_KMTQ · 2022-10-25

**Confidence:** 4
**Correctness:** 4
**Technical Novelty And Significance:** 3
**Empirical Novelty And Significance:** 4
**Recommendation:** 8

**Clarity, Quality, Novelty And Reproducibility:**

The clarity could be improved by making the loss function more explicit and the meaning of the examples trained just to improve the classification robustness (with *NO* IT information) in Fig. 4 (orange dots?).

The quality of the work (topic and good combination of original experimental data and well engineered methods) is remarkable, and the results are novel and interesting.

I expect that the authors makes the recordings and associated stimuli public so that the community can use these input-output pairs in the future.

**Strength And Weaknesses:**

* Strengths:

(a) The understanding of the abilities of biological neural networks in a language that can directly be incorporated in the imporvement of artificial networks is a fundamental topic for the ICLR audience.

(b) The work uses original experimental results from IT (that I guess will be shared with the community) to improve nets previously fitted with general-purpose BrainScore, making them more suited to classification. And they succesfully show that the nets are not only closer to IT (as expected from the loss function), but they also improve the explanation of certain behavioral behavior and the classification performance.

(c) This works represents an advance in the literature that deals with the comparison between artificial and biological networks.

* Weaknesses:

(a) The authors make a really interesting claim that I dont see clearly supported by the results (or I'd like to see it better stressed). In the last paragraph of the introduction and in the first paragraph of the discussion they say that increasing robustness to adversarial attacks does not necessarily imply being more close to IT. This is very interesting from the neuroscience point of view because solving the eventual problems of artificial networks in a specific task (such as classification) does not necessarily make them more human. This implies that eventhough the classification goal may be functionally sensible, the architecture or the actual strategies used by the artificial nets to get the goal are so different from the biological mechanisms that improving the performance of the artificial nets does not do them "more biological", and hence they provide little insight on what the brain may actually be doing.
I see that Figs. 4 and 5 show that improved similarity with IT leads to better robustness in classification (blue points or blue lines, respectively), but in Fig. 5 I dont see how similarity with IT is measured in the orange curve. And in Fig. 4 it is not clear to me how the orange dots were trained: were they trained enforcing robustness to adversarial attacs but not imposing alignments with IT? (is this what "random IT" means?).

(b) Reproduction of behavior is limited to a single experiment. This is fine for a conference paper but the authors should acknowledge that a single experiment does not summarize the rich aspects of visual psychophysics [Bowers et al. BioArxiv. 21]. In this regard, there is literature that proposes that biologically sensible alternatives to the current artificial neurons in conventional artificial architectures improve robustness to adversarial attacks [Bertalmio et al. Sci. Rep. 20] (in line to what is proposed in this work), but also reproduce a range of classical psychophysical (behavioral) results on brightness and texture perception. Similarly, Gomez-Villa et al. Vis.Res. 20 and Li et al. J.Vision 22, show that simpler (more human) architectures better reproduce classical color illusions and the Contrast Sensitivity Functions. The need for more exhaustive connections between physiology/architecture and behavior (wider range of psychophysics) should be acknowledged in the introduction or discussion.

(c) I think readers would appreciate a mathematical expression for the "CKA" loss function rather than the verbal descrioption given in section 2.4

* References:

[Bowers et al. 22] Deep Problems with Neural Network Models of Human Vision
https://psyarxiv.com/5zf4s/

[Bertalmio et al. 20] Evidence for the intrinsically nonlinear nature of receptive fields in vision. Sci.Rep.
https://www.nature.com/articles/s41598-020-73113-0

[Gomez-Villa et al. 20] Color illusions also deceive CNNs for low-level vision tasks: Analysis and implications. Vision Research
https://www.sciencedirect.com/science/article/pii/S0042698920301243

[Li et al. 22] Contrast sensitivity functions in autoencoders. Journal of Vision
https://jov.arvojournals.org/article.aspx?articleid=2778843

**Summary Of The Paper:**

The authors analyze how a better reproduction of brain responses related to object recognition in artificial nets (1) improves their robustness to adversarial attacks, and (2) makes them more aligned with human behavior in object recognition.

The authors propose retraining of classification networks to enforece alignment with the response of the IT region of brain primates. The correspondence between the network and experimental measurements in IT is enforced by using the Centered Kernel Alignment (CKA).
The authors start from CORnet-S [Kubilius et al.19], but, in principle, the proposed loss function could be applied to other classification nets.

The results support the above points (1 and 2) when classifying objects used in the physiological experiments and using one behavioral metric proposed in [Rahalingham et al. 18].


**Summary Of The Review:**

I enjoyed this convincing report showing the improvement of the performance of classification networks when one enforces stronger alignment with biology (and in particular with original IT responses). Limitations are acknowledged and solutions for future physiological experiments are suggested.
As stated above, some clarification in the loss function and in the description of some results is desirable, as well as the acknowledgement of the need for wider check with visual psychophysics (the physiology/architecture-behavior connection is critical), but I definitely think that this work should be presented in ICLR.

---

> ### Author Response · Authors · 2022-11-11
> **Updated evidence for adversarial training claims and anonymous code and data release**
>
> Thank you Reviewer KMTQ for your favorable review of our paper; we were pleased you found our methods and results to be high quality, convincing, and relevant to the ICLR audience. Below we address the concerns you have raised, which we think will help improve the quality of our paper.
>
> Regarding evidence for the claim that ”increasing robustness to adversarial attacks does not necessarily imply being more close to IT,” we thank the reviewer for spotting this mistake on our end; support for this important claim came along late in project, and we now realize Figure 4 was not updated to show networks that were adversarially fine tuned for HVM classification but not fit to IT. [Here](https://imgur.com/a/SVSaCEn) is a draft figure, demonstrating our claim that (CORnet-S adversarially trained on HVM images show in lavender) indeed increases on adversarial accuracy (y-axis), but only marginally improves on IT similarity (x-axis); on the other hand, an IT-fit CORnet-S is both more IT-aligned and more adversarially robust, indicating a different route to robustness than that achieved by adversarial training. We will update our final paper with this figure.
>
> Regarding the orange points on figure 4, we see how this was confusing given that orange is used for adversarially trained networks on figure 5. For clarity, orange points on figure 4 were not adversarially trained; in figure 4, “Random IT” is intended to mean these networks were trained with the standard HVM classification loss as well a CKA loss term encouraging the network to match a simulated response matrix where instead of real IT responses, the stimuli are paired with random Gaussian vectors. We will update the figure 4 text to make this more clear, and to change the color scheme of figure 5, so that there is no visual association between adversarial training and a Random IT CKA penalty.
>
> We agree with the reviewer regarding the limitations of our behavioral experiment, and will add a note as suggested to the discussion emphasizing that this is only one way of comparing with primate behavior, and that more work must be done to validate the effects of our approach on the many other facets of visual psychophysics. We will also add a mathematical expression for CKA to the methods section 2.4. Finally, we have made an anonymous preliminary release of our source code and data [here](https://anonymous.4open.science/r/IT-Alignment-7820/README.md), which we will further polish and document upon release of our final paper.
>
> Again we thank you for your constructive feedback which we believe will improve the clarity of our paper.

---

### Comment · Area_Chair_AbUg · 2022-12-09
**Related work?**

Forgive the late comment -- can the authors speak to the similarity of their method to this work: https://arxiv.org/abs/2209.02582

The method uses CCA to align CNNs to macaque V1 and so seems very related

---

### Decision · Program_Chairs · 2023-01-20

**Decision:**

Accept: notable-top-5%

**Justification For Why Not Higher Score:**

n/a

**Justification For Why Not Lower Score:**

This is a huge amount of work, it's hard to overstate how much work went into this paper.  The evaluation is also really well though out and well motivated.  This paper is technically very well done, and also could be of interest to the ICLR audience as a whole because it ties representations back to their biological origins.

**Metareview: Summary, Strengths And Weaknesses:**

This paper summarizes a model that, in becoming better aligned with macaque IT, becomes more adversarially robust and a closer match to certain human judgements for object recognition. Though the effects are small, this paper provides strong evidence that more robust models are truly more brain-like, because it's a specific tuning of the model towards the brain's representations.  This paper also represents a significant amount of experimental work, which may not be readily apparent to those outside of brain imaging.

The reviewers were unanimously positive about this paper.  The recognized the huge amount of work involved, and the originality of the evaluation approach.  There were questions about a few technical details, but these were largely resolved during review.

**Note From Pc:**

if the above contains the word "oral" or "spotlight" please see: "oral" presentation means -> notable-top-5% and "spotlight" means -> notable-top-25%. As stated in our emails, we are disassociating presentation type from AC recommendations

**Summary Of Ac-Reviewer Meeting:**

n/a